# *ITGAM* is a risk factor to systemic lupus erythematosus and possibly a protection factor to rheumatoid arthritis in patients from Mexico

Julian Ramírez-Bello[1], Celi Sun[2], Guillermo Valencia-Pacheco[3], Bhupinder Singh[2], Rosa Elda Barbosa-Cobos[4], Miguel A. Saavedra[5], Ricardo F. López-Villanueva[6], Swapan K. Nath[2]*

1 Research Unit, Hospital Juárez de México, Mexico City, Mexico, 2 Arthritis and Clinical Immunology Research Program, Oklahoma Medical Research Foundation, Oklahoma City, Oklahoma, United States of America, 3 Hematology Laboratory, Regional Research Center, Autonomous University of Yucatan, Yucatan, Mexico, 4 Rheumatology Department, Hospital Juárez de México, Mexico City, Mexico, 5 Rheumatology Department, Centro Médico Nacional "La Raza", Mexico City, Mexico, 6 Rheumatology Department, Regional Hospital General (ISSSTE), Health Service Yucatan, Yucatan, Mexico

* Swapan-Nath@omrf.org

**Data Availability Statement:** All the summary level data for this manuscript are provided in the main and supplementary Tables.

## Abstract

### Introduction

*ITGAM* has consistently been associated with susceptibility to systemic lupus erythematosus (SLE) in many ethnically diverse populations. However, in populations with higher Amerindian ancestry (like Yucatan) or highly admixed population (like Mexican), *ITGAM* has seldom been evaluated (except few studies where patients with childhood-onset SLE were included). In addition, *ITGAM* has seldom been evaluated in patients with rheumatoid arthritis (RA). Here, we evaluated whether four single nucleotide polymorphisms (SNPs), located within *ITGAM*, were associated with SLE and RA susceptibility in patients from Mexico.

### Methods

Our study consisted of 1,462 individuals, which included 363 patients with SLE (292 from Central Mexico and 71 from Yucatan), and 621 healthy controls (504 from Central Mexico and 117 from Yucatan). Our study also included 478 patients with RA from Central Mexico. TaqMan assays were used to obtain the genotypes of the four SNPs: rs34572943 (G/A), rs1143679 (G/A), rs9888739 (C/T), and rs1143683 (C/T). We also verified the genotypes by Sanger sequencing. Fisher's exact test and permutation test were employed to evaluate the distribution of genotype, allele, and haplotype between patients and controls.

### Results

Our data show that all four *ITGAM* SNPs are significantly associated with susceptibility to SLE using both genotypic and allelic association tests (corrected for multiple testing, but not for population stratification). A second study carried out in patients from Yucatan, a

**Funding:** This work was supported by the National Institute of Health grants (AR60366, MD007909). The funder had no role in study design, data collection and analysis, decision to publish, or preparation of the manuscript.

**Competing interests:** The authors have declared that no competing interests exist.

southeastern part of Mexico (with a high Amerindian ancestry), also replicated SLE association with all four SNPs, including the functional SNP, rs1143679 (OR = 24.6 and $p$ = 9.3X10$^{-6}$). On the other hand, none of the four SNPs are significant in RA after multiple testing. Interestingly, the G**A**CC haplotype, which carries the *ITGAM* rs1143679 (A) minor allele, showed an association with protection against RA (OR = 0.14 and $p$ = 3.0x10$^{-4}$).

## Conclusion

Our data displayed that *ITGAM* is a risk factor to SLE in individuals of Mexican population. Concurrently, a risk haplotype in *ITGAM* confers protection against RA.

## Introduction

*ITGAM* encodes the α-subunit (known as CD11b) of the β2 integrin (known as Mac1 or CD11b/CD18), which has consistently been associated with susceptibility to systemic lupus erythematosus (SLE). In 2008, two genome-wide association studies (GWAS) showed a strong association signal between different single nucleotide polymorphisms (SNPs) located within *ITGAM* and susceptibility to SLE in European-derived populations [1,2]. Another study in 2008, based on a candidate gene (trans-ethnic mapping) approach, pin-pointed a non-synonymous coding variant (rs1143679) within 3$^{rd}$ exon of the *ITGAM*, strongly associated with SLE in African and European-derived populations [3]. *In silico* analysis predicted that this variant, which changed from arginine to histidine at position 77 (R77H), affects the structure and function of this protein [3]. The risk allele of rs1143679 (A allele) has also been associated with transcript and protein levels that are significantly decreased compared to the non-risk allele (G) [4].

Subsequent studies carried out in different African, Asian, Hispanic, and European-derived populations replicated this finding [5–15]. Recently, two GWAS, which included Mexican patients with SLE, also identified to *ITGAM* strongly associated with susceptibility [14,15]. Other candidate gene studies also identified an association between *ITGAM* R77H and SLE in Mexican patients [5,9]. It is important to note that these three studies included childhood-onset SLE patients [5,14,15]. Han *et al.*, for example, included in their study a group of childhood-onset SLE patients from Central Mexico and their results showed the lowest statistical significance in comparison with the rest of the populations evaluated [5]. Several SLE-susceptibility *loci* are differentially-associated between adults and childhood-onset SLE patients; e.g. the functional *PTPN22* R620W polymorphism is a risk factor to pediatric SLE patients from Mexico [16]; however, in this same population, this variant is not associated with adult SLE (identified by two GWAS and three candidate gene studies) [9,14,15,17,18]. On the other hand, Sánchez *et al.* also included a small group of patients with SLE (n = 101) who were from Guadalajara, Morelia, Culiacan, and Mexico City, as well as another group (n = 373) where the great majority of them were of Mexican ancestry born or living in the USA [9]. Guadalajara and Morelia are located in Western Mexico; meanwhile, Culiacan is located in northern Mexico. These places are well known for having a high European ancestry (except Mexico City) and low Amerindian ancestry [19,20]. However, in populations with high Amerindian ancestry, like Yucatan (located southeast of Mexico) [20], *ITGAM* has not been evaluated. Regarding rheumatoid arthritis (RA), the *ITGAM* R77H SNP has seldom been evaluated and the finding of association identified to SLE has not been replicated in this autoimmune disease (AD) [21–26]. Thus, the aim of our study was to evaluate whether *ITGAM* confers

 

susceptibility to adult SLE in a group of individuals from Central Mexico and from Yucatan. We also evaluated these same *ITGAM* SNPs in a group of patients with RA from Central Mexico.

## Material and methods

### Study population

Our study included two groups of SLE cases and controls. The first group was from Central Mexico and included 292 patients with SLE and 504 controls. A second group of SLE cases and controls from Yucatan was included to validate our results. This group consisted of 71 patients with SLE and 117 controls. All the SLE patients were classified using the 1997 American College of Rheumatology (ACR) criteria. To identify a possible role of *ITGAM* SNPs in RA susceptibility, we also included 478 patients from Central Mexico. RA patients were classified using the 2010 ACR-EULAR criteria. Both cases and controls were women older than 18 years of age. All patients and controls were Mexican mestizos who were recruited at the Hospital Juárez de México (HJM), Centro Médico Nacional "La Raza", and at the Hospital General Regional, Servicios de Salud de Yucatán, México. Controls were individuals with no familiar antecedent of AD or inflammatory diseases, including obesity, type 2 diabetes, hypertension, chronic urticarial, allergy to food, asthma, etc., for three generations. This protocol was conducted according to the ethics' guidelines of the Declaration of Helsinki and was approved by the ethics, research, and biosecurity committees of Hospital Juárez de México (HJM 0446/18-I). Written informed consent forms were signed by all participants.

### Isolation of DNA of cases and controls

A total of 6–8 mL of peripheral blood was obtained from the patients and controls into tubes containing EDTA as anticoagulant. Next, the plasma was removed and the buffy coat was obtained to start the extraction of the nuclear DNA using a slightly modified version of the standard method, which involves proteinase K digestion and salification [27]. Finally, the nuclear DNA was quantified and diluted to 5 ng/μL before starting with genotyping.

### Genotyping of *ITGAM* SNPs

We analyzed the *ITGAM* rs1143679G/A (R77H), rs1143683C/T (A858V), rs34572943C/T, and rs9888739C/T SNPs, which were determined by an allelic discrimination assay and TaqMan probes in a CFX96 Touch™ Real-Time Polymerase Chain Reaction (PCR) Detection System (Bio-Rad, Hercules, California, USA), according to the manufacturer's instructions. The PCR reaction for each individual contained 10 ng of DNA, 2.5 μl of TaqMan Master Mix (Applied Biosystems, Foster City, California, USA), 0.065 μl of 40× assay mixture (Applied Biosystems, Foster City, California, USA), and 2.435 μl of DNase free water in a final volume of 5 μl. The PCR conditions used for amplification were as follows: Pre-PCR (1 cycle) 50˚C for 2 minutes and 95˚C for 8 minutes, followed by 45 cycles of denaturing at 95˚C for 15 seconds, and annealing and extension at 60˚C for 1 minute. We genotyped 35% of the samples twice and the reproducibility was 100%.

### Validation of SNPs by Sanger DNA sequencing

To verify the minor alleles at the SNPs in the samples detected by the TaqMan genotyping for the two coding SNPs (rs1143679, rs1143683), we confirmed those alleles by Sanger sequencing. In brief, a subset of (19–21) DNA samples were selected where the SNP shows the presence of minor allele by TaqMan genotype data; these DNA samples were PCR amplified

and sequenced. The genomic DNA was amplified using primers specific to SNP regions (S1 Table). PCR reactions of 10 μl volume containing 20 ng of genomic DNA, 0.5 μM each of forward and reverse primers, 0.1 mM dNTPs, 1x PCR buffer, and 0.2 unit of Taq DNA polymerase was performed in a thermal cycler. The cycling conditions involved initial denaturation at 94˚C for 4 minutes, followed by 35 cycles of denaturation at 94˚C for 1 minute, primer annealing at 55–60˚C for 1 minute, and primer extension at 72˚C for 1 minute. A final extension at 72˚C for 7 minutes was performed and products stored at 4˚C until electrophoresis. The PCR products were resolved by electrophoresis in 2% agarose gels in 1x TBE buffer and visualized by ethidium bromide staining. The PCR amplicons were sequenced using Sanger sequencing. All the sequences along with reference sequence were assembled using the SeqMan tool available in DNAstar software package (www.dnastar.com). We confirmed the presence of minor allele among the DNA samples, which previously showed its presence by genotyping.

## Statistical analysis

For each SNP, we assessed Hardy-Weinberg (H-W) proportion to confirm the independent segregation of the alleles, especially in controls. The minor allele frequencies (MAF) and distribution of genotypes in cases and controls were compared using chi-square test or Fisher's exact test, wherever applicable. For each SNP, both allele-based as well as genotype-based associations were performed using PLINK software [28]. Odds ratio (OR) and 95% confidence interval (CI) were calculated for each SNP. P-values were corrected for multiple testing using Bonferroni correction. We also assessed the SNP-wise significance empirically by using 100,000 permutation-based tests (where the case-control labels were randomly shuffled).

The statistical power of our study was determined using the Quanto software (http://hydra.usc.edu/gxe), which takes into account a Log-additive genetic model, the proportion of cases-controls, the MAF of rs1143679G/A, an OR of 1.5, the prevalence of SLE and RA in Mexicans [29], and the sample size. Haplotype analysis were obtained by using the Haploview software V 4.2 [30].

To perform the meta-analysis, we used METAL software, where it can combine either (a) test statistics and standard errors or (b) p-values across studies (taking sample size and direction of effect into account). (http://csg.sph.umich.edu/abecasis/metal/).

# Results

## Hardy-Weinberg equilibrium and statistical power

None of the SNPs showed any statistical deviation from the H-W proportion. However, the MAF of all four SNPs are slightly different from the control samples from Central Mexico and Yucatan (Tables 1 and 2). The statistical power of our study was 94.4% and 26.4% to patients with SLE from Central Mexico and Yucatan, respectively, meanwhile to RA was 99.3%.

## Association with SLE

At each of the four *ITGAM* SNPs, we first assessed the allelic association between controls and adult SLE patients from Central Mexico. Both nonsynonymous *ITGAM* SNPs were significantly associated: rs1143679 (R77H): allele G vs A: OR 1.65 and $p = 0.005$, and rs1143683 (A858V): C vs T: OR 1.72 and $p = 0.002$, respectively (Table 1). Given the previous association identified in patients with SLE from Central Mexico, we used another group of patients with SLE from Yucatan (who have a high Amerindian ancestry). Our results also showed a strong association with R77H: allele G vs A: OR 24.6 and $p = 9.3 \times 10^{-6}$, and A858V: allele C vs T: OR 6.4 and $p = 5.8 \times 10^{-5}$ (Table 2). In fact, all four SNPs remained significant in both SLE samples, even after accounting for multiple correction. Thus, our results suggest that, independently of

**Table 1. Genotypic and allelic frequencies of the *ITGAM* polymorphisms and association analysis in patients with SLE and controls from Central México.**

| SNP | Position | Genotypic Association | | | | Allelic Association | | | | |
|---|---|---|---|---|---|---|---|---|---|---|
| | | Structure | Case | Cont | P-val | Allele | Case | Cont | OR (95% CI) | P-val |
| | | | (%) | (%) | | | (%) | (%) | | |
| rs34572943 | 31272353 | AA | 3 | 0 | $2.2 \times 10^{-3}$ | A | 61 | 61 | 1.81 (1.25–2.62) | $1.49 \times 10^{-3}$ |
| | | | (1.0) | (0) | | | (10.4) | (6.1) | | |
| | | AG | 55 | 61 | | G | 523 | 947 | | |
| | | | (18.8) | (12.1) | | | (89.6) | (93.9) | | |
| | | GG | 234 | 443 | | | | | | |
| | | | (80.2) | (87.9) | | | | | | |
| rs1143679 | 31276811 | AA | 5 | 2 | $2.9 \times 10^{-2}$ | A | 66 | 71 | 1.65 (1.16–2.34) | $5.19 \times 10^{-3}$ |
| | | | (1.7) | (0.4) | | | (11.3) | (7.2) | | |
| | | AG | 56 | 67 | | G | 518 | 917 | | |
| | | | (19.2) | (13.6) | | | (88.7) | (92.8) | | |
| | | GG | 231 | 425 | | | | | | |
| | | | (79.1) | (86.0) | | | | | | |
| rs9888739 | 31313253 | TT | 6 | 0 | $8.6 \times 10^{-4}$ | T | 71 | 77 | 1.67 (1.91–2.35) | $2.77 \times 10^{-3}$ |
| | | | (2.1) | (0) | | | (12.2) | (7.6) | | |
| | | TC | 59 | 77 | | C | 513 | 931 | | |
| | | | (20.2) | (15.3) | | | (87.8) | (92.4) | | |
| | | CC | 227 | 427 | | | | | | |
| | | | (77.7) | (84.7) | | | | | | |
| rs1143683 | 31336888 | TT | 7 | 1 | $1.7 \times 10^{-3}$ | T | 71 | 76 | 1.72 (1.23–2.43) | $1.62 \times 10^{-3}$ |
| | | | (2.5) | (0.2) | | | (12.6) | (7.7) | | |
| | | TC | 57 | 74 | | C | 491 | 906 | | |
| | | | (20.3) | (15.1) | | | (87.4) | (92.3) | | |
| | | CC | 217 | 416 | | | | | | |
| | | | (77.2) | (84.7) | | | | | | |

Bonferroni corrected p-value for significance = 0.0125

the ancestry component, *ITGAM* is a risk factor to adult SLE patients from Central and Southeast Mexico. In addition, under the dominant genetic model, we also identified an association (Tables 1 and 2).

Using METAL, we performed a meta-analysis, accounting for sample sizes and direction of the ORs. As expected, the minor allele frequencies and ORs are consistent with the directionality, although there is a significant heterogeneity of ORs (S2 Table).

## Association with RA

We used a group of patients with RA to identify the possible role of the four *ITGAM* SNPs in the susceptibility for this AD. Our results did not show an association between *ITGAM* and RA (Table 3). However, it is important to note that both SLE risk alleles at R77H and A858V showed a tendency toward association with protection against RA: OR = 0.73 and $p$ = 0.09 and OR = 0.72 and $p$ = 0.08, respectively. Interestingly, the genotype-based tests were also showing similar tendency (Table 3).

## Haplotype association in SLE and RA

We identified five and four haplotypes in the group of patients with SLE from Central and Southeast México, respectively (Table 4). In both groups of patients, the AATT haplotype,

**Table 2. Genotypic and allelic frequencies of the *ITGAM* polymorphisms and association analysis in patients with SLE and controls from Yucatán.**

| SNP | Position | Genotypic Association | | | | Allelic Association | | | | |
|---|---|---|---|---|---|---|---|---|---|---|
| | | Structure | Case | Cont | P-val | Allele | Case | Cont | OR (95% CI) | P-val |
| | | | (%) | (%) | | | (%) | (%) | | |
| rs34572943 | 31272353 | AA | 2 | 0 | $5.7 \times 10^{-6}$ | A | 19 | 2 | 17.9 (4.11–78.19) | $2.9 \times 10^{-7}$ |
| | | | (2.8) | (0) | | | (13.4) | (0.9) | | |
| | | AG | 15 | 2 | | G | 123 | 232 | | |
| | | | (21.1) | (1.7) | | | (86.6) | (99.1) | | |
| | | GG | 54 | 115 | | | | | | |
| | | | (76.1) | (98.3) | | | | | | |
| rs1143679 | 31276811 | AA | 2 | 0 | $2.6 \times 10^{-4}$ | A | 13 | 1 | 24.6 (3.18–190.3) | $9.3 \times 10^{-6}$ |
| | | | (3.0 | (0) | | | (9.7) | (0.4) | | |
| | | AG | 9 | 1 | | G | 121 | 229 | | |
| | | | (13.4) | (0.9) | | | (90.3) | (99.6) | | |
| | | GG | 56 | 114 | | | | | | |
| | | | (83.6) | (99.1) | | | | | | |
| rs9888739 | 31313253 | TT | 3 | 0 | $4.3 \times 10^{-4}$ | T | 21 | 7 | 5.6 (2.33–13.62) | $2.4 \times 10^{-5}$ |
| | | | (4.2) | (0) | | | (14.8) | (3.0) | | |
| | | TC | 15 | 7 | | C | 121 | 227 | | |
| | | | (21.1) | (6.0) | | | (85.2) | (97.0) | | |
| | | CC | 53 | 110 | | | | | | |
| | | | (74.7) | (94.0) | | | | | | |
| rs1143683 | 31336888 | TT | 1 | 1 | $9.45 \times 10^{-4}$ | T | 17 | 5 | 6.4 (2.32–17.87) | $5.8 \times 10^{-5}$ |
| | | | (1.4) | (0.9) | | | (12.3) | (2.1) | | |
| | | TC | 15 | 3 | | C | 121 | 229 | | |
| | | | (21.7) | (2.6) | | | (87.7) | (97.9) | | |
| | | CC | 53 | 113 | | | | | | |
| | | | (76.8) | (96.6) | | | | | | |

Bonferroni corrected p-value for significance = 0.0125

which carries the minor alleles of the rs34572943, rs1143679, rs9888739, and rs1143679 SNPs showed stronger haplotype association with SLE susceptibility even with 100,000 permutations or Bonferroni correction for multiple testing (Table 4). Regarding RA, we also identified five haplotypes (Table 5). Interestingly, the G**A**CC haplotype, which carries the rs1143679 risk allele (**A**), showed an association with protection against RA (OR = 0.14, *p* = 0.0003, *pc* = 0.0027).

## Discussion

Previous GWA or candidate gene studies have identified various SLE/RA-susceptibility *loci* [1–15,31–33]. *ITGAM* has consistently been associated with SLE susceptibility [1–15]; however, this association has not been observed in other ADs like RA [21–26]. Thus, this gene is well known to be one of the main SLE-susceptibility *loci*. *ITGAM* encodes CD11b, a component of the macrophage-1 antigen complex (Mac1, also known as complement receptor 3 [CR3]), which together with CD18, form Mac-1 or CR3, a protein that mediates leukocyte adhesion, migration, and phagocytosis in different cells including neutrophils, monocytes, macrophages, and dendritic cells [4,34–36]. CD11b also contributes to the phagocytosis of opsonized particles, including apoptotic cells and immune complex [34].

**Table 3. Genotypic and allelic frequencies of the *ITGAM* polymorphisms and association analysis in patients with RA and controls from Central México.**

| SNP | Position | Genotypic Association | | | | Allelic Association | | | | |
|---|---|---|---|---|---|---|---|---|---|---|
| | | Structure | Case (%) | Cont (%) | P-val | Allele | Case (%) | Cont (%) | OR (95% CI) | P-val |
| rs34572943 | 31272353 | AA | 0 (0) | 0 (0) | NA | A | 44 (4.6) | 61 (6.1) | 0.75(0.50–1.12) | $1.5 \times 10^{-1}$ |
| | | AG | 44 (9.2) | 61 (12.1) | | G | 912 (95.4) | 947 (93.9) | | |
| | | GG | 434 (90.8) | 443 (87.9) | | | | | | |
| rs1143679 | 31276811 | AA | 2 (0.4) | 2 (0.4) | $1.9 \times 10^{-1}$ | A | 50 (5.3) | 71 (7.2) | 0.73(0.50–1.06) | $9.3 \times 10^{-2}$ |
| | | AG | 46 (9.8) | 67 (13.6) | | G | 888 (94.7) | 917 (92.8) | | |
| | | GG | 421 (89.8) | 425 (86.0) | | | | | | |
| rs9888739 | 31313253 | TT | 2 (0.4) | 0 (0) | $4.6 \times 10^{-2}$ | T | 56 (5.3) | 77 (7.6) | 0.75 (0.53–1.07) | $1.2 \times 10^{-1}$ |
| | | TC | 52 (10.9) | 77 (15.3) | | C | 900 (94.7) | 931 (92.4) | | |
| | | CC | 424 (88.7) | 427 (84.7) | | | | | | |
| rs1143683 | 31336888 | TT | 3 (0.7) | 1 (0.2) | $4.1 \times 10^{-2}$ | T | 51 (5.7) | 19 (2.2) | 0.72 (0.50–1.04) | $8.0 \times 10^{-2}$ |
| | | TC | 45 (10.1) | 74 (15.1) | | C | 843 (94.3) | 849 (97.8) | | |
| | | CC | 399 (89.2) | 416 (84.7) | | | | | | |

Bonferroni corrected p-value for significance = 0.0125

Different studies have shown that the *ITGAM* rs1143679G/A SNP, which changes from arginine (CGC) to histidine (CAC) at position 77 (R77H), affects the structure and function of this protein, and that the *ITGAM* rs1143679A allele has been associated with transcript and

**Table 4. *ITGAM* haplotype frequencies and association analysis in patients with SLE and controls from Mexico City and Yucatan.** The order of the SNPs is: rs34572943-rs1143679-rs9888739-rs1143683.

| Population | Haplotype | SLE (%) | Controls (%) | OR | 95% CI | Fisher's P | *Pc* |
|---|---|---|---|---|---|---|---|
| Mexico City | GGCC | **85.0** | **87.9** | 0.74 | 0.53–1.02 | 0.06 | 0.49 |
| | AATT | **9.6** | **4.0** | 2.532 | 1.65–3.88 | $1.17 \times 10^{-5}$ | $2.1 \times 10^{-5}$ |
| | GACC | **0.7** | **2.3** | 0.307 | 0.105–0.89 | 0.02 | $3.3 \times 10^{-2}$ |
| | GGCT | **1.4** | **1.4** | 1.019 | 0.42–2.45 | 0.96 | 1.0 |
| | GGTC | **1.3** | **1.2** | 1.085 | 0.42–2.79 | 0.86 | 0.99 |
| Yucatan | GGCC | 82.9 | 95.7 | 0.207 | 0.09–0.46 | $4.17 \times 10^{-5}$ | $2.1 \times 10^{-5}$ |
| | AATT | 8.4 | 0.4 | 21.187 | 3.8–118.10 | $6.74 \times 10^{-5}$ | $7.3 \times 10^{-5}$ |
| | GGCT | 3.2 | 1.3 | 2.52 | 0.56–11.3 | 0.21 | 0.69 |
| | GGTC | 0.8 | 2.2 | 0.35 | 0.05–2.15 | 0.41 | 0.97 |

OR = odds ratio. CI, confidence interval. *pc*: corrected *p*-value after 100,000 permutations.

**Table 5. *ITGAM* haplotype frequencies and association analysis in patients with RA and controls from Central Mexico.** The order of the SNPs is: rs34572943-rs1143679-rs9888739-rs1143683.

| Haplotype | RA (%) | Controls (%) | OR | 95% CI | Fisher's P | *pc* |
|---|---|---|---|---|---|---|
| GGCC | 92.2 | 87.9 | 1.68 | 1.19–2.38 | $2\times10^{-3}$ | $2.8\times10^{-3}$ |
| AATT | 4.0 | 4.0 | 0.98 | 0.61–1.57 | 0.94 | 1.0 |
| GACC | 0.3 | 2.3 | 0.14 | 0.04–0.49 | $3\times10^{-4}$ | $2.7\times10^{-3}$ |
| GGCT | 1.0 | 1.4 | 0.73 | 0.31–1.7 | 0.46 | 0.94 |
| GGTC | 0.8 | 1.2 | 0.68 | 0.26–1.76 | 0.43 | 0.99 |

OR = odds ratio. CI, confidence interval. *pc*: corrected *p*-value after 100,000 permutations.

protein levels significantly decreased compared to the non-risk allele (G) [3,4]. On the other hand, CD11b-mediated phagocytosis has been identified to be reduced in neutrophils from donors carrying the *ITGAM* rs1143679A allele. Thus, this variant impairs many functions of CR3 including a reduction in phagocytosis [34,35]. In 2009, we identified several *ITGAM* SNPs associated with SLE in Mexican patients; however, in this study we included patients with childhood-onset SLE. Notably, in this same population, it has been identified that some genetic risk factors are different in patients with adult SLE and childhood-onset SLE. For example, the functional *PTPN22* C1858T (R620W) variant is not a risk factor to patients with adult SLE (this finding has been reported in five articles; two GWAS and three candidate gene studies) [9,14,15,17,18]; however, this variant represents a risk factor to childhood-onset SLE [16]. In addition, two other articles showed that the *TNFα*-308G/A polymorphism in Mexican populations is not a risk factor to adult SLE [37,38]; however, this variant showed an association with susceptibility to childhood-onset SLE [39]. Two additional studies have identified an association between the *ITGAM* R77H SNP and patients with adult SLE and childhood-onset SLE from Mexico [14–15]. On the other hand, a third study, which also included Mexican adult SLE patients, identified an association with *ITGAM* R77H [9]; however, this report included samples that were predominantly of Mexican ancestry either born or living in the USA (373 patients), and 101 more samples of patients from Mexico City, Guadalajara, Culiacan, and Morelia [9].

Mexico is a country formed by a great population admixture, so each state or region has a different ancestral component. For example, in a study carried out in Mexico City (using autosomal ancestry-informative markers [AIMs]), it was identified that the individuals had 50% Amerindian, 45% Caucasian, and 5% African ancestry [40]. Meanwhile, individuals from Guadalajara and Morelia (both located in Western Mexico), and Culiacan (located in Northern Mexico) have a high percentage of European ancestry and a low percentage of Amerindian ancestry [19,20]. In another population like Yucatan (located in Southeastern Mexico), which presents a high Amerindian ancestry [20], this SLE-susceptibility *locus* has not been analyzed. Thus, to validate and confirm the role of *ITGAM* in the SLE susceptibility in Mexican patients, we included two groups of patients: one of them from central Mexico and another from southeast Mexico (Yucatan). Our data suggest that the *ITGAM* R77H variant is a risk factor to SLE from Mexico City and Yucatan. It is important to comment that, independently of the ancestry in Mexico, our data showed that *ITGAM* is a risk factor to SLE. In our study, we also consider the genotyping call errors, especially for the rs1143679 and rs1143683. To rule out this possibility, we also verified the TaqMan genotyping data by Sanger sequencing, and our genotyping data matched very well (S3 Table). However, our results should be taken with caution because of the small sample size of SLE patients from Yucatan. Additionally, another limitation of our study is, by design, the lack of ability to assess and correct for population stratification.

Therefore, to minimize the effect of population stratification, we have used the ethnicity-matched cases and controls from the Central Mexico and Yucatan. Further studies are warranted to establish this extraordinary *ITGAM*–SLE association in Yucatan.

On the other hand, *ITGAM* rs1143679 (R77H) has seldom been evaluated in patients with RA and the results in these populations have shown that this *locus* is not a risk factor to RA [20–26]. Because there are few studies in this AD (two in Colombians and four in European-derived populations), we evaluated four variants of *ITGAM* in Mexican patients with RA. Our results showed no association between *ITGAM* R77H and RA; however, we identified a trend towards an association (Table 3). Conversely, the G**A**CC haplotype, which represents the combination of the *ITGAM* rs34572943G, rs1143679**A**, rs9888739C, and rs1143679C alleles, showed an association with protection against RA even after correction with the Bonferroni test (OR = 0.14, *p* = 0.0003). To note, this haplotype carries the *ITGAM* rs1143679A minor allele. Thus, our data suggests that *ITGAM* R77H in combination with other variants confers protection to RA. Thus, this is the first study documenting an association between *ITGAM* R77H and protection against RA. However, our results should be evaluated in other populations, including Latin-Americans, to validate our findings. We also acknowledge that our study has several limitations, including the absence of AIMs, and a statistically low powered group of patients with SLE in Yucatan, among others.

In summary, this study validated previous findings in our population: *ITGAM* is a risk factor to adult SLE in a group of individuals in Mexican population. This finding was replicated with a stronger effect in patients from Yucatan (a region with high Amerindian component). In addition, this is the first study documenting an association between an *ITGAM* haplotype (GACC), which carries the *ITGAM* functional (maybe causal for SLE) allele ("A", 77H) of rs1143679 and protection against RA.

## Supporting information

**S1 Table. Primer sequences were used to amplify two SNPs.**
(DOCX)

**S2 Table. Meta-analysis of four ITGAM SNPs with SLE across two cohorts.**
(DOCX)

**S3 Table. Comparison of SNP alleles between TaqMan and Sanger sequencing.**
(DOCX)

## Acknowledgments

These authors are grateful to the patients and their families for their cooperation and blood samples. Also, we would like to thank Ms. Louise Williamson for technical assistance.

## Author Contributions

**Conceptualization:** Julian Ramírez-Bello, Swapan K. Nath.

**Data curation:** Guillermo Valencia-Pacheco, Rosa Elda Barbosa-Cobos, Miguel A. Saavedra, Ricardo F. López-Villanueva, Swapan K. Nath.

**Formal analysis:** Julian Ramírez-Bello, Celi Sun, Bhupinder Singh.

**Funding acquisition:** Swapan K. Nath.

**Supervision:** Swapan K. Nath.

**Writing – original draft:** Julian Ramírez-Bello, Swapan K. Nath.

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
