## [Editor Report · Decision Letter 0]

30 Aug 2019

PONE-D-19-23844

ITGAM is a risk factor to systemic lupus erythematosus and possibly a protection factor to rheumatoid arthritis in patients from Mexico

PLOS ONE

Dear Dr. Nath,

Thank you for submitting your manuscript to PLOS ONE. After careful consideration, we feel that it has merit but does not fully meet PLOS ONE’s publication criteria as it currently stands. Therefore, we invite you to submit a revised version of the manuscript that addresses the points raised during the review process.

We would appreciate receiving your revised manuscript by Oct 14 2019 11:59PM. To enhance the reproducibility of your results, we recommend that if applicable you deposit your laboratory protocols in protocols.io, where a protocol can be assigned its own identifier (DOI) such that it can be cited independently in the future. For instructions see: http://journals.plos.org/plosone/s/submission-guidelines#loc-laboratory-protocols

We look forward to receiving your revised manuscript.

Kind regards,

Amr H Sawalha

Academic Editor

PLOS ONE

Journal Requirements:

a) Please provide an amended Funding Statement that declares *all* the funding or sources of support received during this specific study (whether external or internal to your organization) as detailed online in our guide for authors at http://journals.plos.org/plosone/s/submit-now.  

b) Please state what role the funders took in the study.  If any authors received a salary from any of your funders, please state which authors and which funder. If the funders had no role, please state: "The funders had no role in study design, data collection and analysis, decision to publish, or preparation of the manuscript."

Additional Editor Comments:

The work by Ramirez-Bello and colleagues evaluate the well-established genetic association between ITGAM and lupus in 2 cohorts from Mexico. They replicate this genetic association in 2 Mexican populations and provide results suggesting a possible protective effect of ITGAM lupus-associated haplotype against RA. The is straight forward work, well written, and replication of this genetic association in Mexican populations is worth reporting.

The authors need to address the following points before this can be accepted for publication:

1) As the authors are well aware, the single most important limitation in this work is the lack of ability, by study design, to assess and correct for population stratification. This is major issue in admixed populations such as in Mexico. This limitation although mentioned briefly in the manuscript, should be emphasized and also mentioned in the abstract. On the positive side, the authors did study patients and controls matched by region (central Mexico versus Yucatan).

2) The association between ITGAM and lupus in the Yucatan population is likely inflated as a result of low power and small sample size. This should be also noted in the abstract. Power has been estimated, and in particular given low power in the Yucatan cohort it is important to attempt to correct the odds ratio for inflation as a function of power similar to an approach used to calculate winner's curse effect.

3) Please attempt to preform a meta-analysis combining both Yucatan and Central Mexico lupus cohort.

4) Other minor points:

- Please amend the first sentence in the conclusion to say "Our data show that all four ITGAM SNPs are significantly associated with

susceptibility to SLE in patients from Mexico"

- Table 1 legend should say "Central Mexico"

- Table 3 RA findings are not from Yucatan but from Central Mexico.
---

## [Author Response · Author response to Decision Letter 0]

11 Oct 2019

Response to the Reviewers comments.

The work by Ramirez-Bello and colleagues evaluates the well-established genetic association between ITGAM and lupus in two cohorts from Mexico. They replicate this genetic association in two Mexican populations and provide results suggesting a possible protective effect of ITGAM lupus-associated haplotype against RA. This is straight-forward work, well written, and replication of this genetic association in Mexican populations is worth reporting.

The authors need to address the following points before this can be accepted for publication:

Thank you for reviewing our manuscript, and here are our point-by-point responses to the comments.

Q1) As the authors are well aware, the single most important limitation in this work is the lack of ability, by study design, to assess and correct for population stratification. This is major issue in admixed populations such as in Mexico. This limitation although mentioned briefly in the manuscript, should be emphasized and also mentioned in the abstract. On the positive side, the authors did study patients and controls matched by region (central Mexico versus Yucatan).

Response: Yes, we agree with the reviewer regarding limitation of this study. As reviewer appreciated that to minimize the effect of population stratification, we have used the ethnicity-matched cases and controls from the Central Mexico and Yucatan. Now we have mentioned this in abstract and in the text.

Q2) The association between ITGAM and lupus in the Yucatan population is likely inflated as a result of low power and small sample size. This should be also noted in the abstract. Power has been estimated, and in particular given low power in the Yucatan cohort it is important to attempt to correct the odds ratio for inflation as a function of power similar to an approach used to calculate winner's curse effect.

Response: Yes, we agree that association results from Yucatan are likely to be inflated due to the small samples sizes, especially due to the low allele frequency, and now we have mentioned this in the abstract. As you suggested, we have performed a meta-analysis, which is adjusted by the samples sizes. As expected, the minor allele frequencies and ORs are in consistent with the directionality, although there is a significant heterogeneity of ORs. We have provided all this information in the Supplementary Table 1.

Q3) Please attempt to perform a meta-analysis combining both Yucatan and Central Mexico lupus cohort.

Response: Yes, we have performed a meta-analysis, and information is provided in the Supplementary Table 3.

4) Other minor points:

- Please amend the first sentence in the conclusion to say: "Our data show that all four ITGAM SNPs are significantly associated with susceptibility to SLE in patients from Mexico."

Response: Done, thank you.

- Table 1 legend should say "Central Mexico"

Response: Corrected, thank you.

- Table 3 RA findings are not from Yucatan but from Central Mexico.

Response: Corrected. Thank you.

---

## [Editor Report · Decision Letter 1]

17 Oct 2019

ITGAM is a risk factor to systemic lupus erythematosus and possibly a protection factor to rheumatoid arthritis in patients from Mexico

PONE-D-19-23844R1

Dear Dr. Nath,

We are pleased to inform you that your manuscript has been judged scientifically suitable for publication and will be formally accepted for publication once it complies with all outstanding technical requirements.

With kind regards,

Amr H Sawalha

Academic Editor

PLOS ONE
---

## [Editor Report · Acceptance letter]

20 Nov 2019

PONE-D-19-23844R1 

*ITGAM* is a risk factor to systemic lupus erythematosus and possibly a protection factor to rheumatoid arthritis in patients from Mexico 

Dear Dr. Nath:

I am pleased to inform you that your manuscript has been deemed suitable for publication in PLOS ONE. Congratulations! Your manuscript is now with our production department. 

With kind regards,

on behalf of

Dr. Amr H Sawalha 

Academic Editor

PLOS ONE